# Replication Study: Coadministration of a tumor-penetrating peptide enhances the efficacy of cancer drugs

**Christine Mantis, Irawati Kandela, Fraser Aird, Reproducibility Project: Cancer Biology\***

Developmental Therapeutics Core, Northwestern University, Evanston, United States

**\*For correspondence:** tim@cos.io; nicole@scienceexchange.com

**Group author details:**
Reproducibility Project: Cancer Biology See page 11

**Abstract** In 2015, as part of the Reproducibility Project: Cancer Biology, we published a Registered Report (Kandela et al., 2015) that described how we intended to replicate selected experiments from the paper "Coadministration of a tumor-penetrating peptide enhances the efficacy of cancer drugs" (Sugahara et al., 2010). Here we report the results of those experiments. We found that coadministration with iRGD peptide did not have an impact on permeability of the chemotherapeutic agent doxorubicin (DOX) in a xenograft model of prostate cancer, whereas the original study reported that it increased the penetrance of this cancer drug (Figure 2B; Sugahara et al., 2010). Further, in mice bearing orthotopic 22Rv1 human prostate tumors, we did not find a statistically significant difference in tumor weight for mice treated with DOX and iRGD compared to DOX alone, whereas the original study reported a decrease in tumor weight when DOX was coadministered with iRGD (Figure 2C; Sugahara et al., 2010). In addition, we did not find a statistically significant difference in TUNEL staining in tumor tissue between mice treated with DOX and iRGD compared to DOX alone, while the original study reported an increase in TUNEL positive staining with iRGD coadministration (Figure 2D; Sugahara et al., 2010). Similar to the original study (Supplemental Figure 9A; Sugahara et al., 2010), we did not observe an impact on mouse body weight with DOX and iRGD treatment. Finally, we report meta-analyses for each result.

## Introduction

The Reproducibility Project: Cancer Biology (RP:CB) is a collaboration between the Center for Open Science and Science Exchange that seeks to address concerns about reproducibility in scientific research by conducting replications of selected experiments from a number of high-profile papers in the field of cancer biology (*Errington et al., 2014*). For each of these papers a Registered Report detailing the proposed experimental designs and protocols for the replications was peer reviewed and published prior to data collection. The present paper is a Replication Study that reports the results of the replication experiments detailed in the Registered Report (*Kandela et al., 2015*) for a 2010 paper by Sugahara et al., and uses a number of approaches to compare the outcomes of the original experiments and the replications.

In 2010, Sugahara et al. reported that a novel cyclized form of a RGD motif containing peptide (iRGD) increased penetrance simply through co-administration with other therapies, including peptide-based therapeutics, small molecule drug compounds, and nanoparticle-based therapeutics (*Sugahara et al., 2010*). This followed-up on a previous study that reported iRGD coupled to a motif that binds to the Neuropilin-1 receptor, increased tissue penetrance of cancer drugs beyond the vasculature when it was directly conjugated to those chemotherapies (*Feron, 2010*; *Sugahara et al., 2009*).

The Registered Report for the 2010 paper by Sugahara et al. described the experiments to be replicated (Figure 2 and Supplemental Figure 9A), and summarized the current evidence for these findings (*Kandela et al., 2015*). Since that publication there have been additional studies that have utilized the iRGD peptide with the chemotherapeutic agent doxorubicin (DOX) in cancer biology research. Follow-up studies by Peng et al. reported that in an *in vitro* model of prostate cancer iRGD improved the penetrance of DOX into tumor cell lines (*Peng and Kopeček, 2015*). However, this improvement was only observed in a DOX/iRGD conjugated form. Further, in an *in vivo* model of mammary adenocarcinoma, Ni et al. reported that iRGD-grafted nanocrystallites can target cancer stem cells, which are typically inside the tumor core (*Ni et al., 2015*). In a hepatocarcinoma xeno-graft model, utilizing the same mouse strain as Sugahara et al., Schmithals et al. reported that co-administration of iRGD enhanced the penetration of DOX into tumor tissue as well as reduced tumor size as compared to DOX alone (*Schmithals et al., 2015*; *Sugahara et al., 2010*). Utilizing a different chemotherapeutic, Zhang et al. reported that in an A549 xenograft model of non-small cell lung cancer, co-administration of iRGD along with gemcitabine lead to increased penetrance of drug to the tumor and decreased tumor volume over the treatment period of 30 days (*Zhang et al., 2015*). Finally, Lao et al. reported that a thymosin alpha 1-iRGD conjugated form increased apoptosis of MCF-7 cells and decreased tumor volume over an 11 day treatment period when compared to thy-mosin alpha 1 alone (*Lao et al., 2015*).

The outcome measures reported in this Replication Study will be aggregated with those from the other Replication Studies to create a dataset that will be examined to provide evidence about repro-ducibility of cancer biology research, and to identify factors that influence reproducibility more generally.

## Results and discussion

### Quantifying the amount of DOX present in tumor tissue and major organs in mice treated with DOX with or without iRGD

We sought to independently replicate the impact iRGD has on the penetrance of DOX, in an uncon-jugated form, in tumor and organ tissues of mice bearing orthotopic 22Rv1 human prostate tumors. This experiment is similar to what was reported in Figure 2B of *Sugahara et al. (2010)*. This experi-ment utilized a 10 mg/kg dose of DOX and analyzed DOX accumulation after a brief exposure (1 hr). Mice harboring tumors were injected with phosphate buffered saline (PBS), 10 mg/kg DOX and PBS, or a combination of 10 mg/kg DOX and 4 μmol/kg iRGD. Tissues from mice were examined for DOX accumulation using matched tissues from mice injected with PBS as the blank reference samples (*Figure 1*, *Figure 1—figure supplement 1*). DOX accumulation in tumors from mice treated with DOX and iRGD was 0.86 times [n=4, SD=0.68] the amount of DOX in tumors from mice treated with DOX and PBS [n=4, M=1.0, SD=0.45]. The comparison of these two groups, which was planned *a priori*, was not statistically significant (Two-tailed Student's *t*-test; $t(6) = 0.352$, $p=0.737$, *a priori* sig-nificance threshold = 0.05). This differed from the ~7.15 times increase in DOX accumulation in tumors from mice treated with DOX and iRGD compared to DOX alone reported in *Sugahara et al. (2010)*.

### Effect of Dox alone or Dox in combination with iRGD on tumor growth and total body weight

To test if iRGD enhances the effect of DOX on tumor growth, mice bearing orthotopic 22Rv1 tumors implanted 2 weeks earlier received intravenous injections every other day of either PBS, 1 mg/kg DOX and PBS (DOX + PBS), or 1 mg/kg DOX combined with 4 μmol/kg of iRGD (DOX + iRGD). This experiment is similar to what was reported in Figure 2C of *Sugahara et al. (2010)*. While the original study included two doses of DOX (1 mg/kg or 3 mg/kg), this replication attempt was restricted to the lower dose (1 mg/kg), which in the original study showed the largest effect when DOX + PBS and DOX + iRGD were compared. The tumors were harvested and weighed 24 days after the start of treatment. PBS treated mice achieved an average tumor weight of 1.20 grams [n=7, SD=0.87], which decreased to an average tumor weight of 0.721 grams [n=7, SD=0.239] in mice treated with DOX + PBS (*Figure 2*). This is similar to the tumor weights reported in *Sugahara et al. (2010)*, which were ~1.2 grams and ~0.82 grams for PBS and DOX + PBS, respectively. However, while the original

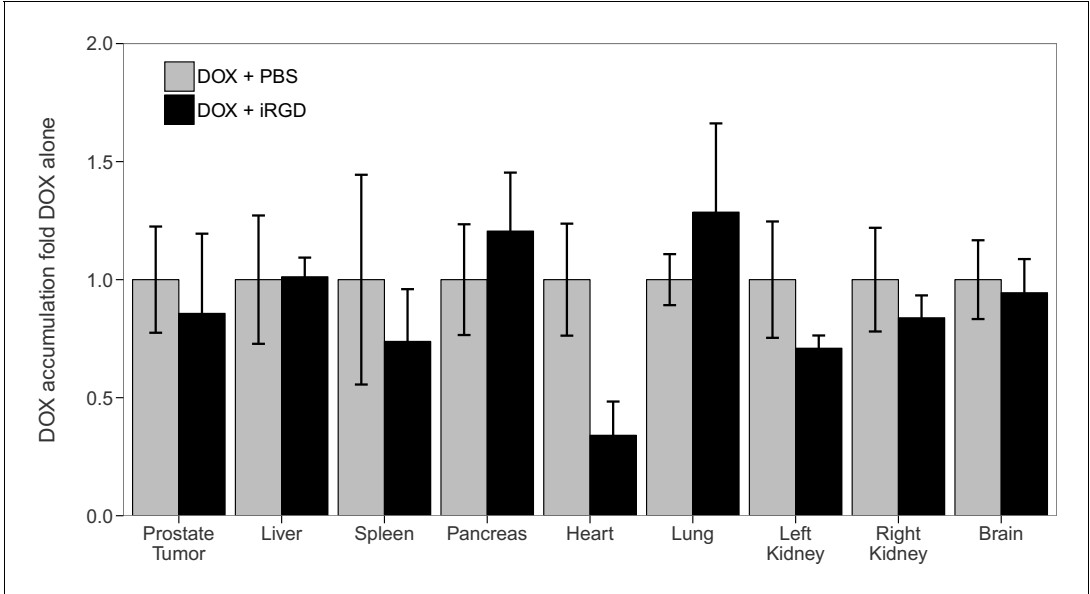

**Figure 1.** Tissue specific DOX accumulation. Mice bearing orthotopic 22Rv1 human prostate tumors were intravenously injected with a mixture of PBS, 10 mg/kg DOX and PBS (DOX + PBS), or 10 mg/kg DOX and 4 µmol/kg of iRGD (DOX + iRGD). One hr later tissues were harvested and DOX was quantified by spectrophotometry using absorbance at 490 nm. Tissues from DOX + PBS and DOX + iRGD treated mice [n=4 for both conditions] were examined for DOX accumulation using matched tissues from mice injected with PBS [n=2] as the blank reference samples. DOX accumulation for each tissue was normalized to the mean absorbance of that same tissue type treated with 10 mg/kg DOX and PBS. Means reported and error bars represent s.e.m. Unpaired two-tailed Student's $t$-test between DOX + PBS and DOX + iRGD for prostate tumor tissue; $t(6) = 0.352$, $p=0.737$, with *a priori* alpha level = 0.05. Additional details for this experiment can be found at https://osf.io/d4zeg/.

The following figure supplement is available for figure 1:

**Figure supplement 1.** This is the same experiment as in *Figure 1*, but with the $OD_{490}$ readings plotted for each condition instead of the $OD_{490}$ relative to DOX + PBS for each tissue.

study reported a final average tumor weight of ~0.35 grams from mice treated with DOX + iRGD, this replication attempt observed a final average tumor weight of 0.668 grams [n=7, *SD*=0.576]. Analysis of tumor weights did not detect any statistically significant differences (One-way ANOVA *F* (2, 18) = 1.58, *p*=0.233). The pairwise comparison of tumor weights from mice treated with DOX + PBS compared to DOX + iRGD was also not statistically significant (Two-tailed Welch's *t*-test; *t*(8.01) = 0.227, *p*=0.826, *a priori* significance threshold = 0.05).

During the course of the treatment, mouse body weights were monitored to evaluate whether DOX given alone or combined with iRGD negatively impacted the health of the animals. This is similar to what was reported in Supplementary Figure 9A of *Sugahara et al. (2010)*, except only the 1 mg/kg dose of DOX conditions were tested. Mice treated with PBS, DOX + PBS, or DOX + iRGD were measured every 4 days throughout the 24 days of treatment (*Figure 3*, *Figure 3—figure supplement 1*). There was no shift in body weight during the course of the treatments. This is similar to the original study, which only saw a shift in body weight when a higher dose (3 mg/kg) of DOX was administered. Analysis on percent body weight shift of all groups on day 24 did not detect any statistically significant differences (One-way ANOVA *F*(2, 18) = 1.666, *p*=0.217), *a priori* significance threshold = 0.05. While we conducted this analysis because it was prespecified in the Registered Report (*Kandela et al., 2015*) and provides a direct comparison to the original analysis, there are additional approaches that could be taken to explore these data. An additional exploratory analysis, as described in the Registered Report (*Kandela et al., 2015*), was performed by calculating the area under the curve (AUC) of body weight for each mouse over the entire treatment period. Similar to the Day 24 end-point analysis, the analysis on AUC of all groups did not detect any statistically significant differences (One-way ANOVA *F*(2, 18) = 1.072, *p*=0.363).

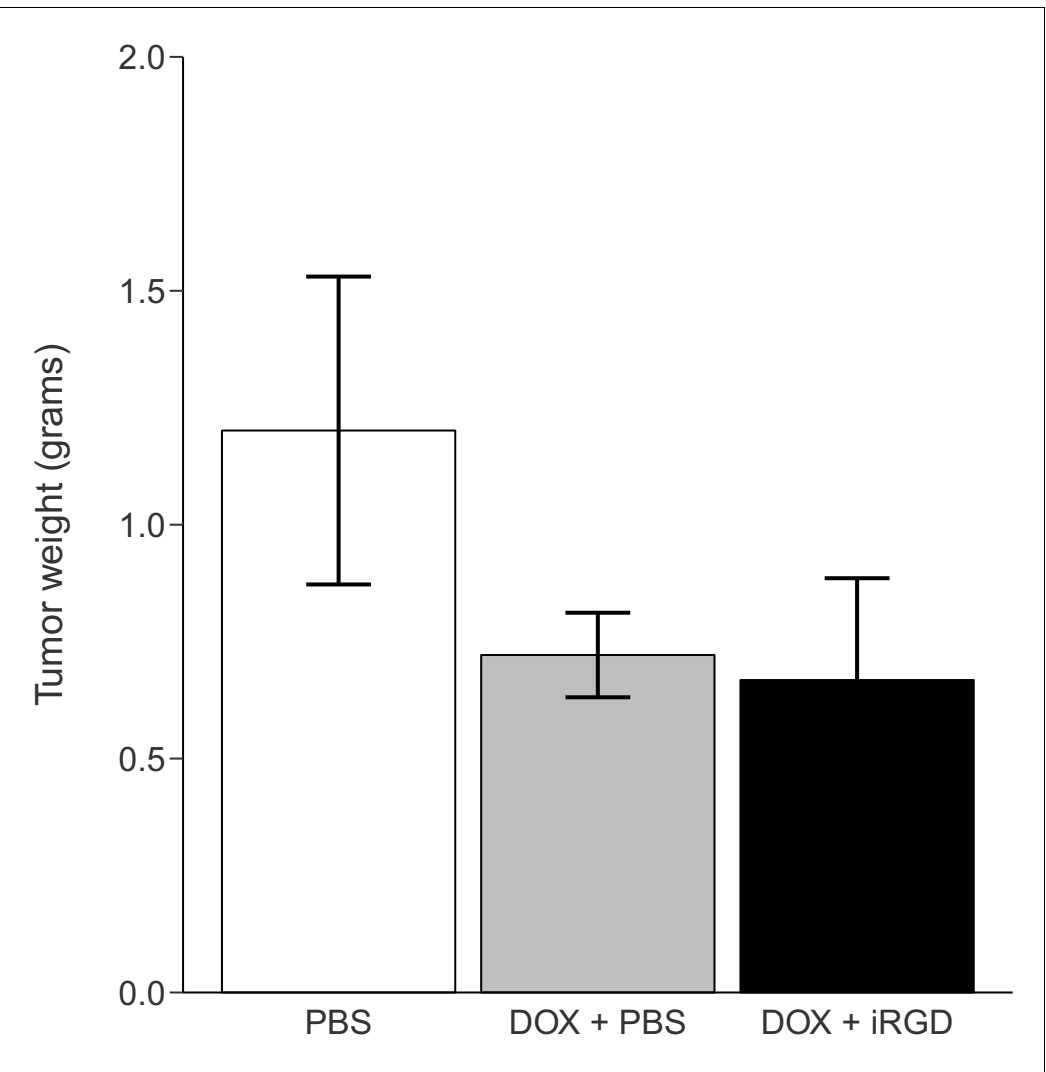

**Figure 2.** Tumor weight following treatment. Mice harboring orthotopic 22Rv1 human prostate tumors were intravenously injected with PBS alone (PBS), 1 mg/kg DOX and PBS (DOX + PBS), or 1 mg/kg DOX and 4 µmol/kg iRGD (DOX + iRGD). Mice were treated every other day for 24 days and 1 hr after the last treatment tumors were harvested and weighed. Means reported and error bars represent s.e.m. Number of mice per condition (n=7; n=21 mice total). One-way ANOVA on tumor weights of all groups; $F_{(2, 18)}$ = 1.58, $p$=0.233. Two-tailed Welch's $t$-test between DOX + PBS and DOX + iRGD; $t_{(8.01)}$ = 0.227, $p$=0.826, with *a priori* alpha level = 0.05. Additional details for this experiment can be found at https://osf.io/kwh39/.

## Assessment of TUNEL staining of tumor and heart tissue after treatment

To further assess the effect of iRGD on DOX efficacy and toxicity, terminal deoxynucleotidyl transferase-mediated deoxyuridine triphosphate nick end labeling (TUNEL) staining, an indicator of cellular apoptosis, was performed (*Figure 4*, *Figure 4—figure supplement 1*). This experiment is similar to what was reported in Figure 2D of *Sugahara et al. (2010)* except that this replication attempt was restricted to the 1 mg/kg DOX conditions. Tumors, to evaluate efficacy, and heart tissues, to evaluate cardiotoxicity a known effect of DOX (*Arola et al., 2000*), were excised after the mice received intravenous injections of PBS, DOX + PBS, or DOX + iRGD every other day over a 24 day period. Negative and positive controls for TUNEL staining were performed in parallel to the study tissues confirming the specificity of the staining (*Figure 4—figure supplement 1B,C*). While TUNEL staining was detectable in tumors (*Figure 4B–D*), no detectable TUNEL staining in heart tissues was

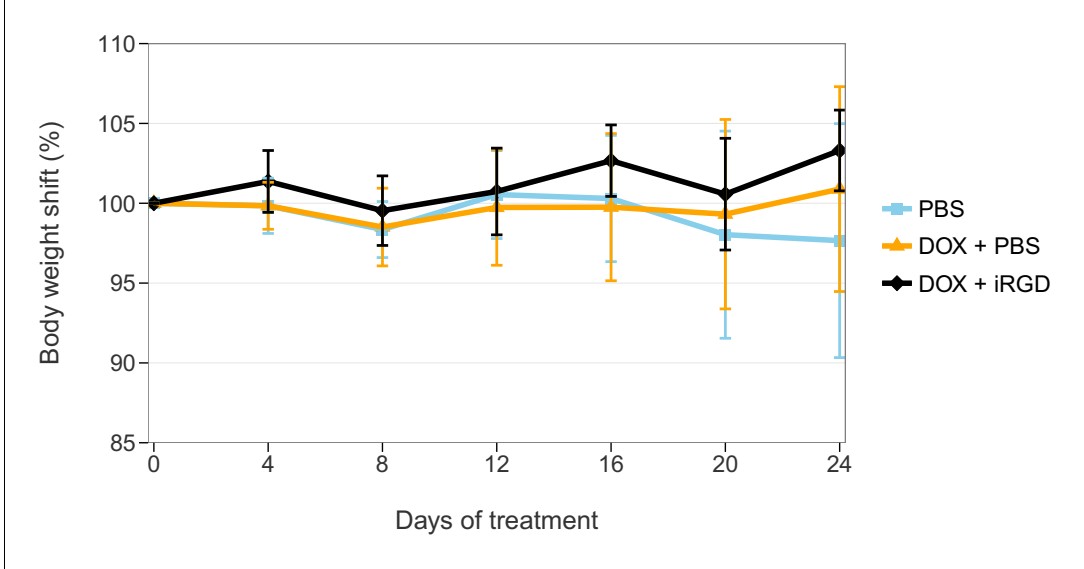

**Figure 3.** Body weight shift of mice during treatment. Mice bearing orthotopic 22Rv1 human prostate tumors were intravenously injected with PBS alone (PBS), 1 mg/kg DOX and PBS (DOX + PBS), or 1 mg/kg DOX and 4 μmol/kg of iRGD (DOX + iRGD). Mice were treated every other day for 24 days with total body weight measured every four days during the treatment. On day 0 body weight was considered 100% for each animal. Means reported and error bars represent *SD*. Number of mice per condition (n=7; n=21 mice total). One-way ANOVA on percent body weight shift of all groups on day 24; $F(2, 18) = 1.666$, $p=0.217$. Additional details for this experiment can be found at https://osf.io/kwh39/.
The following figure supplement is available for figure 3:

**Figure supplement 1.** Total body weight during treatment.

observed (*Figure 4E–G*). This is in contrast to the original study, which reported TUNEL positive scores in heart tissue for these conditions, albeit much smaller than the reported scores in heart tissues from the 3 mg/kg DOX conditions. Scoring of TUNEL positive cells in tumors from mice treated with DOX + PBS were 0.79 times [n=6, *SD*=0.35] the amount of DOX in tumors from mice treated with PBS alone [n=6, *M*=1.0, *SD*=0.37] (*Figure 4A*). Similarly, tumors from mice treated with DOX + iRGD were 0.71 times [n=6, *SD*=0.20] the amount of DOX in tumors from PBS treated mice. This is in comparison to the original study, which reported ~1.4 times increase for DOX + PBS treated mice and an ~2.6 times increase in TUNEL positive cells for DOX + iRGD relative to untreated mice. To evaluate if there were any differences in TUNEL staining in tumors among the conditions, we performed an ANOVA, which was not statistically significant (One-way ANOVA $F(2, 15) = 1.378$, $p=0.282$). The planned pairwise comparison of TUNEL staining in tumors from mice treated with DOX + PBS compared to DOX + iRGD was also not statistically significant ($t(15) = 0.435$, $p=0.670$, *a priori* significance threshold = 0.05). Other studies that utilized a similarly low dose of doxorubicin to reduce side effects, also reported small reductions in tumor weight compared to vehicle control with minimal and not statistically significant changes in TUNEL staining (*Hossain et al., 2012*; *Sugahara et al., 2015*; *Wang et al., 2010*).

## Meta-analyses of original and replicated effects

We performed a meta-analysis using a random-effects model to combine each of the effects described above as pre-specified in the confirmatory analysis plan (*Kandela et al., 2015*). To provide a standardized measure of the effect, a common effect size was calculated for each effect from the original and replication studies. Glass' Δ is the standardized difference between two means using the standard deviation of only the control group. It is used in this case because of the unequal variance between the control and treatment conditions in the original and replication studies. The effect size $r$ is a standardized measure of the strength and direction of the association between two

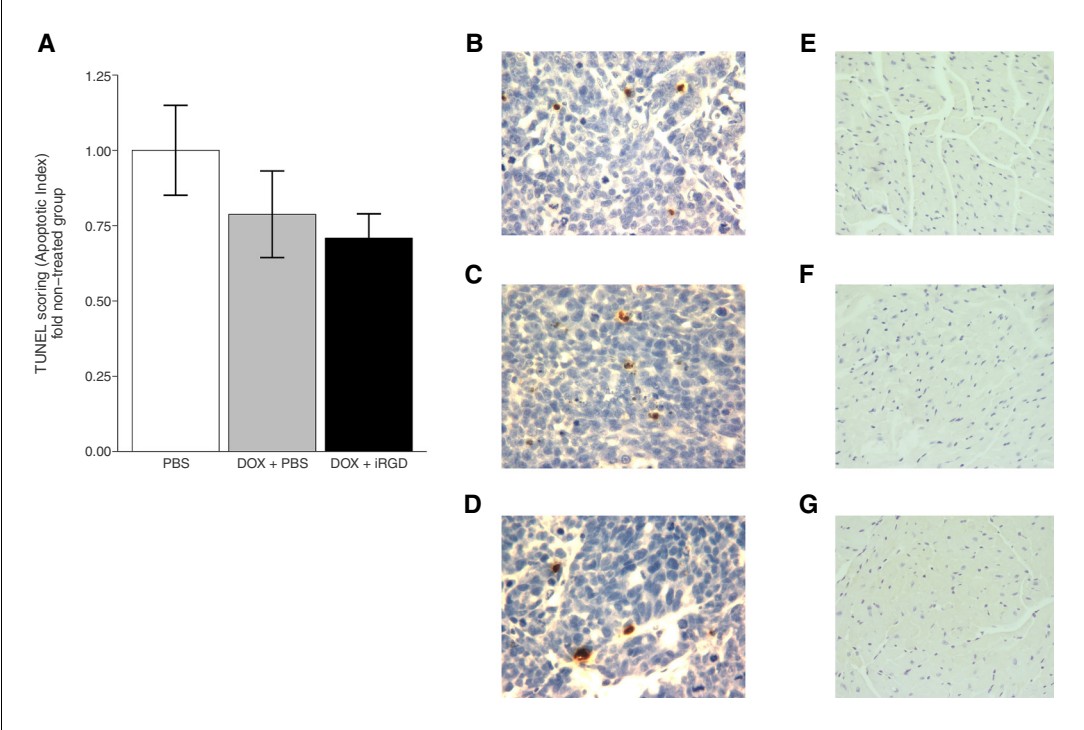

**Figure 4.** TUNEL staining of mouse tissues. Mice bearing orthotopic 22Rv1 human prostate tumors were intravenously injected with PBS alone (PBS), 1 mg/kg DOX and PBS (DOX + PBS), or 1 mg/kg DOX and 4 μmol/kg of iRGD (DOX + iRGD). TUNEL staining was performed on tumor and heart sections of each animal. (**A**) Boxplot of mean apoptotic index calculated from TUNEL stained tumor sections. TUNEL scores were normalized to the average score of tumors from PBS treated mice. Means reported and error bars represent s.e.m. Number of mice per condition (n=6; n=18 mice total). One-way ANOVA on apoptotic index of all groups; $F(2, 15) = 1.378$, $p=0.282$. Planned contrast between DOX + PBS and DOX + iRGD; $t(15) = 0.435$, $p=0.670$ with *a priori* alpha level = 0.05. Representative images of TUNEL staining of tumor sections from PBS (**B**), DOX + PBS (**C**), or DOX + iRGD (**D**) treated mice. Representative images of TUNEL staining of heart sections from PBS (**E**), DOX + PBS (**F**), or DOX + iRGD (**G**) treated mice. Additional details for this experiment can be found at https://osf.io/7eynw/.

The following figure supplement is available for figure 4:

**Figure supplement 1.** This is the same experiment as in *Figure 4*, but with the apoptotic index plotted for each condition instead of the apoptotic index relative to PBS treated tumors.

variables. In this case, the treatment condition and the dependent variable (i.e. body weight). Cohen's *d* is the standardized difference between two means using the pooled sample standard deviation.

The comparison of DOX accumulation in tumors from 10 mg/kg DOX and PBS treated mice compared to 10 mg/kg DOX and 4 μmol/kg iRGD treated mice resulted in Glass' Δ = 7.41, 95% CI [0.89, 14.38] for the data estimated *a priori* from *Sugahara et al. (2010)*, *Figure 2B*. This compares to Glass' Δ = −0.32, 95% CI [−1.70, 1.11] reported in this study. In both calculations the standard deviation of the DOX + PBS control group was used because of unequal variance in the original study. A meta-analysis (*Figure 5A*) of these two effects resulted in Glass' Δ = 4.44, 95% CI [−2.92, 11.81], $p=0.237$. The replication and the original results are in opposite directions when considering the effect, and the point estimate of the replication effect size was not within the confidence interval of the original result, or vice versa. Also, the random effects meta-analysis did not result in a statistically significant effect. Further, the Cochran's Q test for heterogeneity was statistically significant ($p=0.0454$), which along with a large confidence interval around the weighted average effect size from the meta-analysis suggests heterogeneity between the original and replication studies.

The comparison of prostate tumor weights from 1 mg/kg DOX and PBS treated mice vs. 1 mg/kg and 4 μmol/kg iRGD treated mice resulted in Glass' Δ = 1.61 with a 95% CI [0.44, 2.73] for the data estimated *a priori* from *Sugahara et al. (2010)*, *Figure 2C*. This compares to Glass' Δ = 0.22 with a

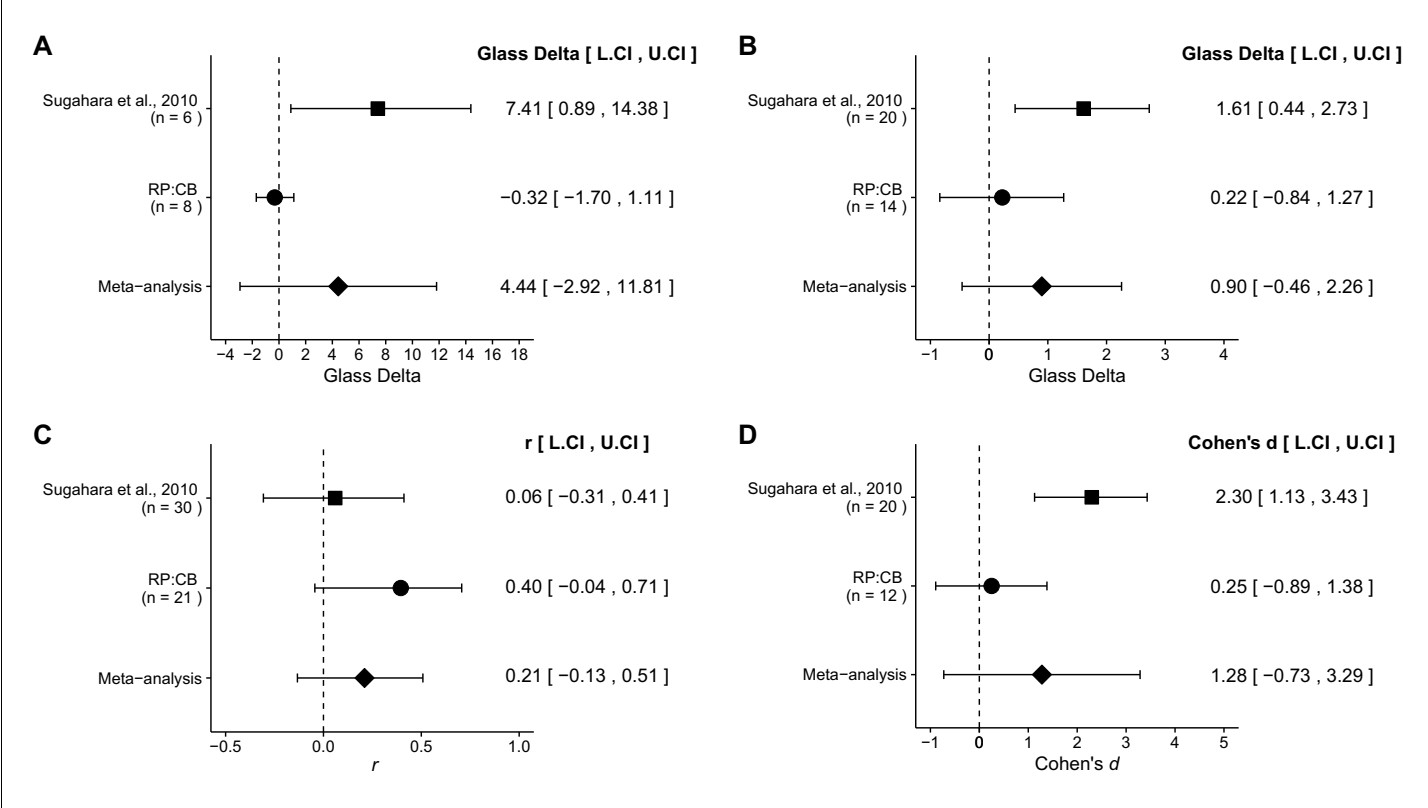

**Figure 5.** Meta-analyses of each effect. Effect size and 95% confidence interval are presented for *Sugahara et al. (2010)*, this replication attempt (RP: CB), and a random effects meta-analysis to combine the two effects. Sample sizes used in *Sugahara et al. (2010)* and this replication attempt are reported under the study name. (**A**) Dox accumulation in tumor tissue of mice treated with 10 mg/kg DOX alone or 10 mg/kg DOX and 4 μmol/kg iRGD (meta-analysis *p*=0.237). (**B**) Tumor weight of mice treated with 1 mg/kg DOX or 1 mg/kg DOX and 4 μmol/kg iRGD (meta-analysis *p*=0.195). (**C**) Body weight shift of mice treated with PBS, 1 mg/kg DOX and PBS, or 1 mg/kg DOX and 4 μmol/kg iRGD (meta-analysis *p*=0.229). (**D**) TUNEL staining from mice treated with 1 mg/kg DOX or 1 mg/kg DOX and 4 μmol/kg iRGD (meta-analysis *p*=0.211). Additional details for these meta-analyses can be found at https://osf.io/ymxaz/.

95% CI [−0.84, 1.27] reported in this study. In both calculations the standard deviation of the DOX + PBS control group was used because of unequal variance in the original and replication studies. A meta-analysis (*Figure 5B*) of these two effects resulted in Glass' Δ = 0.90 with a 95% CI [−0.46, 2.26], *p*=0.195. Both results are consistent when considering the direction of the effect, however the point estimate of the replication effect size was not within the confidence interval of the original result, or vice versa. The random effects meta-analysis did not result in a statistically significant effect.

The comparison of body weight shift at day 24 from mice treated with PBS, DOX + PBS, or DOX + iRGD resulted in *r* = 0.06, 95% CI [−0.31, 0.41] for the data estimated *a priori* from *Sugahara et al. (2010)*, Supplemental Figure 9A. This compares to *r* = 0.40, 95% CI [−0.04, 0.71] reported in this study. A meta-analysis (*Figure 5C*) of these two effects resulted in *r* = 0.21 95% CI [−0.13, 0.51], *p*=0.229. Neither the original nor the replication results were statistically significant. The effect seen in the original and the replication were in the same direction and the point estimate of the original effect size was within the confidence interval of the replication result, and the point estimate of the replication effect size was within the confidence interval of the original result. The random effects meta-analysis did not result in a statistically significant effect suggesting body weight does not change based on treatment in either the original or replication studies.

The comparison of cellular apoptosis, as determined by a TUNEL assay, from 1 mg/kg DOX and PBS treated mice compared to 1 mg/kg and 4 μmol/kg iRGD treated mice resulted in Cohen's *d* = 2.30, 95% CI [1.13, 3.43] for the data estimation *a priori* from *Sugahara et al. (2010)*, *Figure 2D*.

This compares to $d$ = 0.25, 95% CI [−0.89, 1.38] reported in this study. A meta-analysis (*Figure 5D*) of these two effects resulted in $d$ = 1.28, 95% CI [−0.73, 3.29], $p$=0.211. Both results are consistent when considering the direction of the effect, however the point estimate of the replication effect size was not within the confidence interval of the original result, and vice versa. The random effects meta-analysis did not result in a statistically significant effect. Further, the Cochran's Q test for heterogeneity was statistically significant ($p$=0.0212), which along with a large confidence interval around the weighted average effect size from the meta-analysis suggests heterogeneity between the original and replication studies.

This direct replication provides an opportunity to understand the present evidence of these effects. Any known differences, including reagents and protocol differences, were identified prior to conducting the experimental work and described in the Registered Report (*Kandela et al., 2015*). However, this is limited to what was obtainable from the original paper, which means there might be particular features of the original experimental protocol that could be critical, but unidentified. So while some aspects, such as number of cells injected, strain of mice, and drug treatment schedule were maintained, others were unknown or not easily controlled for. These include variables such as the microbiome of recipient mice (*Macpherson and McCoy, 2015*), housing temperature in mouse facilities (*Kokolus et al., 2013*), cell line drift (*Hughes et al., 2007*), circadian biological responses to therapy (*Fu and Kettner, 2013*), and differing compound potency resulting from different stock solutions or differences in peptide synthesis (*Kannt and Wieland, 2016*). Whether these or other factors influence the outcomes of this study is open to hypothesizing and further investigation, which is facilitated by direct replications and transparent reporting.

## Materials and methods

As described in the Registered Report (*Kandela et al., 2015*), we attempted a replication of the experiments reported in Figures 2B-D and Supplemental Figure 9A of *Sugahara et al. (2010)*. A detailed description of all protocols can be found in the Registered Report (*Kandela et al., 2015*). Additional detailed experimental notes, data, and analysis are available on the Open Science Framework (OSF) (RRID: SCR_003238) (https://osf.io/xu1g2/; *Mantis et al., 2016*).

### Peptide synthesis

iRGD: H-Cys-Arg-Gly-Asp-Lys-Gly-Pro-Asp-Cys-NH2 disulfide bridge: C1-C9 The iRGD peptide was chemically synthesized using Fmoc (9-fluorenylmethoxy carbonyl) chemistry by LifeTein, LLC (Somerset, New Jersey). The peptide chains were synthesized from the carboxyl terminus to the Cys amino terminus onto H-Cys(Trt)-2Cl resin. This H-Cys(Trt)-2Cl resin was incubated with dichloromethane (DCM) for 30 min and then washed with dimethylformamide (DMF) three times. Fmoc-protecting groups at the amino terminus were deprotected with an alkaline buffer and then washed with DMF three times to remove the deprotection buffer. The second amino acid was Fmoc-Asp(Otbu)-OH coupled to the first amino acid and then DMF cleaned. After each coupling, the peptide was ninhydrin tested and the coupling and washing steps repeated until the crude peptide was fully synthesized. The full synthesized crude peptide was cleaved from the resin and cleaned with Trifluoroacetic acid (TFA). The crude peptides were diethyl ether precipitated, drained and washed. The rest of the Fmoc group was removed. The peptides were isolated and purified by high-performance liquid chromatography (HPLC) (Shimadzu LC-20A) (data available at https://osf.io/m58sv/). Fractions of greater than 95% purity were used for the investigation. The purity and molecular weight of the respective peptides were confirmed by matrix-assisted laser desorption ionization (MALDI)-time of flight (TOF) mass spectrometry (Shimadzu SEC(GPC)-MALDI-TOF-MS) (data available at https://osf.io/mjxkv/). Detailed synthesis protocols available at (https://osf.io/k9zu3/).

### Cell culture

22Rv1 prostate cells (ATCC, CRL-2505) were maintained in DMEM with 10% fetal bovine serum and penicillin/streptomycin at 37°C/5% $CO_2$. Quality control data for the 22Rv1 cell lines are available at (https://osf.io/en6ru/). This includes results confirming the cell lines were free of Mycoplasma contamination and common mouse pathogens. Additionally, STR DNA profiling of the cell lines was performed and all cells were confirmed to be the indicated cell lines when queried against STR profile databases.

## Animals

All animal procedures were approved by the Northwestern University IACUC# IS00000556 and were in accordance with the Northwestern University's policies on the care, welfare, and treatment of laboratory animals. Blinding occurred during TUNEL analysis. For all experiments five-week old male Athymic Nude mice (Harlan Laboratories, HSD: Athymic Nude-Foxn1nu, Order code 069(nu)/070(nu/+)) were used with the exception of 8 week old male C57BL/6J mice (Jackson Laboratory, RRID: IMSR_JAX:000664) that were used as a positive controls in the TUNEL assay. Athymic Nude mice were inoculated with 22rv1 prostate cancer cells at a density of $1x10^6$ cells in 10 µl of Dulbecco's Phosphate Buffered Saline (PBS) (Sigma-Aldrich, cat # D8537) into the ventral prostate glands of mice. Orthotopic tumors were allowed to grow for two weeks before mice were randomized and assigned to experiments. Each cage contained up to five mice and offered Certified Rodent Diet (Harlan Teklad, cat # 7912) and water *ad libitum*. The animal room was set to maintain between 68–75°F, a relative humidity of 30–70%, a minimum of 15 room air changes per hour, and a 12 hr light/dark cycle, which was interrupted for study-related activities.

## Dose preparation

Weighed compounds were dissolved in PBS. Doxorubicin-HCl (Sigma-Aldrich, cat # D1515) was prepared at a final concentration of 5.335 mg/mL for dose administration of 10 mg/kg in tumor and organ DOX penetrance analysis or 0.533 mg/mL for dose administration of 1 mg/kg in multiday experimentation. The iRGD Peptide was prepared at a concentration of 1.896 mg/mL for dose administration of 4 µmol/kg for all experiments.

## Dose administration

Mice were intravenously injected (IV) by tail vein, based on body weight on injection day. For tumor and organ penetrance analysis, *Figure 1*, mice were anesthetized, perfused, and sacrificed 1 hr after drug administration. For multi-day experiments mice were injected by IV every other day for 24 days based on body weight on injection day. On the last day, mice were anesthetized, perfused, and sacrificed 1 hr after drug administration. Further details of these methods are available at (https://osf.io/bkhnp/).

## Clinical observation

Animals were checked twice after drug administration (AM and PM) for mortality, abnormalities, and signs of pain or distress. Detailed observation was conducted two times after injection and 24 hr post dosing.

## Body weight measurement

For tumor and organ DOX penetrance analysis, body weight was recorded once on injection day by balance (Ohaus Corp. USA, model # Scout Pro SP202). For multiday experiments, body weight was recorded every four days, for 24 days, with additional reads taken on injection days to properly administer compound dose. The treatment group was not blinded to the scientist. Weights taken for each mouse are available at (https://osf.io/y82jp/).

## Perfusion method

Procedure was performed as outlined in the Registered Report (*Kandela et al., 2015*). Briefly, 1 hr after the last dose administration animals were anesthetized with isoflurane, placed on a heating pad, and perfused through the heart with DMEM + 1% BSA using an 18G x 3/4in needle (Terumo Winged Infusion Set# SV18BLK). Only animals that organs were pale after perfusion were taken for further analysis.

## Tumor and organ DOX penetrance analysis

Organs (Prostate Tumor, Liver, Spleen, Pancreas, Heart, Lung, Kidneys, and Brain) were collected 1 hr after treatment according to the perfusion protocol, as described in the Registered Report (*Kandela et al., 2015*), and weight recorded on scale (Mettler Toledo model # XL-300). 100 mg of each organ was homogenized separately in 0.5 mL solution of 1% sodium dodecyl sulfate and 1mM $H_2SO_4$ in water (pH 7.4) using mortar and pestle. A total of 2 mL of chloroform: isopropyl alcohol

(1:1, v/v) was added in stepwise manner, 0.5 mL each time for 4 times. Next, the ground tissue was pipetted several times in order to collect the entire sample from the mortar and placed in a 5 mL Eppendorf tube. A freeze thaw protocol was next used. Samples were placed on dry ice for 1 min followed by a 5 min thaw in a 25°C water bath. Samples were spun at 14,000xg for 15 min at RT and the organic phase was collected and stored at 4°C. Absorbance was read on a UV Vis Spectrophotometer at $OD_{490}$ (Perkin Elmer Lambda 650 UV/VIS Spectrometer #L6020031). The control untreated samples were matched appropriately (e.g. liver for liver and tumor for tumor) to create the blank reference for each tissue and the samples were read against these respective blanks and $OD_{490}$ values were recorded.

## Multi-day dosing protocol

After 24 days, the last dose was administered and 1 hr later all surviving animals were sacrificed according to the perfusion protocol, as described in the Registered Report (*Kandela et al., 2015*). Hearts and prostate tumors were collected and whole organ weights (g) were recorded on scale. Tissues were placed in 4% Paraformaldehyde overnight at 4°C and cut in half. One half was submitted for sectioning to the Mouse Histology Phenotyping Laboratory, Northwestern University (MHPL).

## Histopathology

Half of each heart and prostate tumor sample, in 4% Paraformaldehyde was paraffin embedded, and sectioned into 7 slices per sample with a 50-micron gap. TUNEL staining was performed on 5 slides and 1 slide was treated as negative control. The positive control was dexamethasone (DEX)-treated C57BL/6J mouse thymus. An 8 week old male C57BL/6J mouse (Jackson Laboratory, RRID: IMSR_JAX:000664) was given an intraperitoneal injection of dexamethasone (0.1 mg/g body weight). After 6 hr the animal was sacrificed and the thymus was removed. The thymus was fixed in 10% buffered neutral formalin solution, dehydrated, and embedded in paraffin.

## TUNEL analysis

TUNEL (EMD Millipore, cat # S7100) stained slides were evaluated by Dr. Gennadiy Bondarenko and Dr. Andrey Ugolkov. Images were captured using a Carl Zeiss Axial Lab A1 microscope and a 40x objective, by Dr. Ugolkov, or a Olympus BX45 microscope and a 40x objective, by Dr. Bondarenko (images available at: https://osf.io/3fs27/). Drs. Bondarenko and Ugolkov were blinded to group allocation, only receiving the animal ID with H or P designation to indicate heart or prostate tumor. The frequency of apoptosis was calculated as an apoptotic index, in which the proportion of cells undergoing apoptosis was expressed as a percentage of all cells observed. The apoptotic index of each tissue sample was calculated as the number of TUNEL-positive cells and bodies per 500 cells/microscopic view or 2500 cells/slide (5 slides/tissue), counted in five randomly selected microscopic fields in each tissue sample. Percent apoptotic index was calculated with the following formulation: (i/500) X100%. i = cell undergoing apoptosis. An average was taken of the apoptotic index from all five fields of all five slices (25 fields total) which is considered one biological replicate. Negative and positive control sections were stained in parallel to the tumor and heart samples and are available at (https://osf.io/gmcyt/). Original counts are available at (https://osf.io/pbg7x/).

## Statistical analysis

Statistical analysis was performed with R software (RRID: SCR_001905), version 3.2.3 (*R Core Team, 2016*). All data csv files and analysis scripts are available at (https://osf.io/xu1g2/). Confirmatory statistical analysis was pre-registered (https://osf.io/9hr2d/) before the experimental work began as outlined in the Registered Report (*Kandela et al., 2015*). Additional exploratory analysis (area under the curve) was performed using the weights of the mice over the treatment period. Data were checked to ensure assumptions of statistical tests were met. A meta-analysis of a common original and replication effect size was performed using a random effects model and the *metafor* R package (*Viechtbauer, 2010*). (available at https://osf.io/ymxaz/). The original study data were extracted *a priori* from the published figures by determining the mean and upper/lower error values for each data point. The extracted data were published in the Registered Report (*Kandela et al., 2015*) and were used in the power calculations to determine the sample sizes for this study. In the meta-

analyses where Glass' $\Delta$ was used, because of unequal variance between the two conditions being compared, the standard deviation of DOX + PBS was used in the calculations.

## Deviations from registered report

The source of the TUNEL kit is different than what is listed in the Registered Report, with the used source and catalog number listed above. (note: the original source was not specified). Additional materials and instrumentation not listed in the Registered Report, but needed during experimentation are also listed.

The Registered Report indicated the TUNEL sections would be analyzed with a Scanscope scanner and ImageJ software, while this replication attempt performed the analysis by blinded counting of TUNEL-positive cells and bodies in random fields as described above. All images are available at https://osf.io/3fs27/.

## Acknowledgements

The Reproducibility Project: Cancer Biology would like to thank the following companies for generously donating reagents to the Reproducibility Project: Cancer Biology; American Type and Tissue Collection (ATCC), Applied Biological Materials, BioLegend, Charles River Laboratories, Corning Incorporated, DDC Medical, EMD Millipore, Harlan Laboratories, LI-COR Biosciences, Mirus Bio, Novus Biologicals, Sigma-Aldrich, Mouse Pathology Core of Northwestern University, and System Biosciences (SBI).

## Additional information

### Group author details

Reproducibility Project: Cancer Biology

Elizabeth Iorns: Science Exchange, Palo Alto, United States; Alexandria Denis: Center for Open Science, Charlottesville, United States; Stephen R Williams: Center for Open Science, Charlottesville, United States; Nicole Perfito: Science Exchange, Palo Alto, United States; Timothy M Errington, http://orcid.org/0000-0002-4959-5143: Center for Open Science, Charlottesville, United States

### Competing interests

CM, IK and FA: Developmental Therapeutics Core is a Science Exchange associated lab. RP:CB: EI, NP: Employed by and hold shares in Science Exchange Inc.

### Funding

| Funder | Author |
| --- | --- |
| Laura and John Arnold Foundation | Reproducibility Project: Cancer Biology |

The funder had no role in study design, data collection and interpretation, or the decision to submit the work for publication.

### Author contributions

CM, IK, FA, Acquisition of data, Drafting or revising the article; RP:CB, Analysis and interpretation of data, Drafting or revising the article

### Ethics

Animal experimentation: All animal procedures were approved by the Northwestern University IACUC# IS00000556 and were in accordance with the Northwestern University's policies on the care, welfare, and treatment of laboratory animals.

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
