## [Decision Letter]

Thank you for submitting your article "Replication Study: Coadministration of a tumor-penetrating peptide enhances the efficacy of cancer drugs" for consideration by *eLife*. Your article has been reviewed by two peer reviewers, and the evaluation has been overseen by a Reviewing Editor and Sean Morrison as the Senior Editor. The reviewers have opted to remain anonymous.

The reviewers have discussed the reviews with one another and the Reviewing Editor has drafted this decision to help you prepare a revised submission.

This paper attempts to replicate the work of Sugahara et al. 2010. The authors performed a well designed analysis of the co-administration approach and conclude that there is no statistical difference between groups of animals treated with doxorubicin with or without the iRGD tumor penetrating peptide. Studies of this nature are extremely important to clarify the potential for errors in the scientific literature and to understand how often this actually happens vs. poorly designed/attempts at repeating the study vs. unsubstantiated innuendo/rumor. Overall, this is a well performed study. The reviewers made a few suggestions for improvement before acceptance.

Essential revisions:

This paper is not written clearly. Given that the main objective is to replicate the work of Sugahara et al. 2010, this paper does not even cite the correct Sugahara paper. There are 2 other Sugahara citations (2009 and 2015), but not the 2010 paper.

The section on TUNEL staining results is very poorly written. This section refers to Figure 4 and Figure 5. But no such figure relevant to TUNEL is in this paper. Although Figure 5 of this paper (Cohen's d for meta analysis) may be relevant to TUNEL, it does not appear to be the case from what the authors write. The entire section on TUNEL staining, therefore, needs to be rewritten more clearly to the reader.

The statistical analyses are, in general, fine. However, the authors need to explain their meta analysis more clearly under the statistical analysis section of the methods. First, for the sake of completeness, it will be good to spell out clearly the sample sizes of each analysis undertaken. Second, explain how data from the Sugahara et al. 2010 paper was extracted for meta analysis. Third, please provide some insights into the meaning of Glass' Δ, r and Cohen's d. For example, it appears that Glass' δ is used because the standard deviation of the Sugahara et al. 2010 paper could not be obtained from that paper. Giving such explanations to motivate the use of each measure presented in Figure 5 will be helpful for the reader.

The authors ought to be appreciated for their efforts in making the data and computer programs available in this replication effort. However, some additional details will be helpful. For example, the computer program does not seem to have details about anova or meta analyses. Further, it is not clear whether the% weights reported in Figure 3 were used as outcomes in anova or whether any transformation of these values were done.

Finally, a minor stylistic comment. In Figure 1, the dark shaded boxes are DOX+iRGD and the light shades are for DOX+PBS. However in the later figures (see, for example, Figure 2), the dark shaded boxes are for PBS and the white shaded box is for DOX+iRGD and the light shade is for DOX. I suggest the authors to use consistent shading across figures. For example, use white shaded box for DOX+iRGS in all box plots etc.

---

## [Author Response]

*Essential revisions:*

*This paper is not written clearly. Given that the main objective is to replicate the work of Sugahara et al. 2010, this paper does not even cite the correct Sugahara paper. There are 2 other Sugahara citations (2009 and 2015), but not the 2010 paper.*

Thank you for catching this oversight. We left out the 2010 paper in the reference section, but have added it in the revised manuscript. We have also thoroughly revised the manuscript.

*The section on TUNEL staining results is very poorly written. This section refers to Figure 4 and Figure 5. But no such figure relevant to TUNEL is in this paper. Although Figure 5 of this paper (Cohen's d for meta analysis) may be relevant to TUNEL, it does not appear to be the case from what the authors write. The entire section on TUNEL staining, therefore, needs to be rewritten more clearly to the reader.*

We have completely revised this section. Additionally, the figures associated with TUNEL staining have been revised.

*The statistical analyses are, in general, fine. However, the authors need to explain their meta analysis more clearly under the statistical analysis section of the methods. First, for the sake of completeness, it will be good to spell out clearly the sample sizes of each analysis undertaken. Second, explain how data from the Sugahara et al. 2010 paper was extracted for meta analysis. Third, please provide some insights into the meaning of Glass' Δ, r and Cohen's d. For example, it appears that Glass' δ is used because the standard deviation of the Sugahara et al. 2010 paper could not be obtained from that paper. Giving such explanations to motivate the use of each measure presented in Figure 5 will be helpful for the reader.*

Thank you for these suggestions. We have included them in the revised manuscript. In the statistical analysis section we have included how the data was extracted from the Sugahara et al. 2010 paper for the meta analyses. We explained the meta-analyses section of the Results/Discussion section to include the definition of each effect size measure. Additionally, in the forest plots we have included the sample size of the original and replication attempt that was used in each meta-analysis.

*The authors ought to be appreciated for their efforts in making the data and computer programs available in this replication effort. However, some additional details will be helpful. For example, the computer program does not seem to have details about anova or meta analyses. Further, it is not clear whether the% weights reported in Figure 3 were used as outcomes in anova or whether any transformation of these values were done.*

We have revised the annotation of where these files are to allow for easier discoverability. The data and analysis scripts for each analysis are broken down by experiment. This includes the meta-analyses, which is separate from the experimental analyses. The meta-analyses script (‘Study 15 meta analyses.R’) is available in the files section of the link provided in the Figure 5 figure legend (https://osf.io/ymxaz/files/). Regarding ANOVA of body weights in Figure 3, we have revised the manuscript to clearly state that the% body weight shift values were used in the ANOVA on day 24 values, while the total body weight was used in the additional AUC analysis (files available in the file section of the link provided in Figure 3 figure legend (https://osf.io/kwh39/files/). We have also added a supplement figure to present the data both ways.

*Finally, a minor stylistic comment. In Figure 1, the dark shaded boxes are DOX+iRGD and the light shades are for DOX+PBS. However in the later figures (see, for example, Figure 2), the dark shaded boxes are for PBS and the white shaded box is for DOX+iRGD and the light shade is for DOX. I suggest the authors to use consistent shading across figures. For example, use white shaded box for DOX+iRGS in all box plots etc.*

Thank you for this suggestion. We have revised the figures to have consistent shading of conditions.